# Sensory Substitution for the Visually Impaired: A Study on the Usability of the Sound of Vision System in Outdoor Environments

Otilia Zvorișteanu [ID], Simona Caraiman *[ID], Robert-Gabriel Lupu, Nicolae Alexandru Botezatu and Adrian Burlacu [ID]

Faculty of Automatic Control and Computer Engineering, "Gheorghe Asachi" Technical University of Iasi, D. Mangeron 27, 700050 Iasi, Romania; otilia.zvoristeanu@academic.tuiasi.ro (O.Z.); robert-gabriel.lupu@academic.tuiasi.ro (R.-G.L.); nicolae-alexandru.botezatu@academic.tuiasi.ro (N.A.B.); adrian.burlacu@academic.tuiasi.ro (A.B.)
* Correspondence: simona.caraiman@academic.tuiasi.ro

**Abstract:** For most visually impaired people, simple tasks such as understanding the environment or moving safely around it represent huge challenges. The Sound of Vision system was designed as a sensory substitution device, based on computer vision techniques, that encodes any environment in a naturalistic representation through audio and haptic feedback. The present paper presents a study on the usability of this system for visually impaired people in relevant environments. The aim of the study is to assess how well the system is able to help the perception and mobility of the visually impaired participants in real life environments and circumstances. The testing scenarios were devised to allow the assessment of the added value of the Sound of Vision system compared to traditional assistive instruments, such as the white cane. Various data were collected during the tests to allow for a better evaluation of the performance: system configuration, completion times, electro-dermal activity, video footage, user feedback. With minimal training, the system could be successfully used in outdoor environments to perform various perception and mobility tasks. The benefit of the Sound of Vision device compared to the white cane was confirmed by the participants and by the evaluation results to consist in: providing early feedback about static and dynamic objects, providing feedback about elevated objects, walls, negative obstacles (e.g., holes in the ground) and signs.

**Keywords:** sound of vision; visually impaired; sensory substitution system; outdoor environments; perception; mobility





## 1. Introduction and Related Work

The World Health Organization (WHO) estimates that at least 2.2 billion persons around the world suffer from blindness or visual impairment. The effects of reduced or absent eyesight have a major impact on the life of a person, e.g., daily routine, school, work. In the last years, several systems were proposed to help visually impaired people to improve their perception and/or navigation in unknown environments. These devices incorporate different technologies and sensors.

Di Mattia et al. [1] proposed a low consumption radar-based system for obstacle avoidance. An acoustic warning is generated every time an obstacle is detected, and the range of detection is within 5 m. To obtain a more complex and accurate estimation of environment objects, a sensor fusion system comprising a low-power millimeter wave (MMW) radar and an RGB-Depth (RGB-D) sensor is described in [2,3]. Using this data fusion, the authors ensured the accuracy and stability of the system under any illumination conditions and expand the object detection range up to 80m. Semantic/non-semantic acoustic feedback is sent to the user by Bluetooth bone conduction headphones. In [4], the authors combined the advantages of an IR sensor with an RGB-D camera. A random

sample consensus (RANSAC) segmentation and surface normal vector estimation are used to detect the traversable area. The device is functional in both indoor and outdoor environments and has been tested by eight visually impaired volunteers. The EyeCane [5] is equipped with IR sensors that capture distance information about obstacles and convey it to the user's hand through vibrations.

In the recent years, multiple computer vision based devices for visually impaired have been proposed in the literature. A Raspberry Pi with a camera module was used by Abraham et al. in [6]. The device can identify and locate specific object from the environment, detect text and convey it to speech and also to determine the walkable area. A neural network, i.e., YOLOv3, was used to compute the elements listed before. A module comprising a pi-camera and a controller to move the camera in the required direction was proposed in [7], and this module was integrated into the white-cane. Mask R-CNN is used to detect and classify the objects from the environment. Furthermore, the system estimates the position of obstacles in outdoor environments. Kang et al. [8] proposed a method to detect the risk of collision in a variety of scenarios. The approach effectively locates obstacles at a risk of collision using the shape variation of a grid, called deformable grid. This solution is further improved in [9] by introducing a vertex deformation function to represent the displacement of each vertex in the deformable grid.

User experience understanding is essential to make assistive technology really useful, non-obtrusive and pervasive. Building a technology for the assistance of the visually impaired (VI) requires a deep user study to iteratively assess user satisfaction and then to bring improvements and corrections to the system accordingly. Offline tests and evaluations of the computer vision techniques employed by these assistive systems are truly required to assess technical performance. However, most of the reported contributions are limited to this form of evaluation [10–16], whereas extensive testing with visually impaired users would bring more insight on usability. There are only a few contributions that report taking the system 'in the wild', i.e., in real-life uncontrolled scenarios, to evaluate its performance concerning the technical design and implementation or usability. The obstacle detection system described by Rodriguez et al. [17] has been tested with visually impaired people in real life scenarios consisting of crowded and uncontrolled areas such as a railway station. Experiments in various uncontrolled environments have also been reported by [18,19]. The framework proposed in [20] uses voice messages to alert the user about the presence of obstacles. The system is evaluated with the help of visually impaired subjects and answers to the following aspects: Are the users able to start the application on their own? Can they safely navigate in a novel environment? Is it possible to avoid obstacles using the set of acoustic warnings? Is the system globally useful and can it complement the white cane? In [21], a smartphone camera was used to acquire images from the environment that are further processed on a server. Four visually impaired persons with partial level of visual impairment tested the solution. A questionnaire regarding the overall impression, user interface and experience and alert frequency was collected.

For most of the assistive solutions proposed in the literature, there is a lack of usability assessment. Such evaluations should be based on a more complex feedback provided through visually impaired user experience. Several development loops followed by user evaluations should be employed before reaching a final solution that provides both technical accuracy and user adoption. More visually impaired user evaluations should be designed to assess each component of the system, ranging from the information required to be extracted from the environment to the method of delivering it, but also the various combinations of these components. Furthermore, none of the analyzed systems employ extensive testing in real environments and in uncontrolled settings. New requirements could emerge from these tests, from both technical and user perspective.

## 2. Purpose of the Study

The Sound of Vision system (SoV) [22–24] is a sensory substitution device (SSD) that allows a visually impaired user to perceive unknown environments and to navigate safely.

It works by permanently scanning the environment, extracting essential features and rendering them to the user through audio and haptic means.

Several design–implementation–evaluation loops have been previously employed for the development of the SoV system. At each development phase, various usability aspects have been carefully assessed: selection of the most appropriate audio and haptic encodings [25–27], the effect of training on performance improvement [28,29], and cognitive and affective assessment of mobility tasks [30]. These previous evaluations have been deployed in controlled laboratory settings.

In contrast, the present study is focused on evaluating the usability of the Sound of Vision system in complex real life environments, outdoors. For this purpose, evaluations to assess user perception and mobility in outdoor environments were devised. The tests were performed in normal lighting (i.e., cloudy to bright sunlight) and weather conditions (i.e., no rain or snow, temperatures above 0 °C). The users are assumed to be familiar with all the encodings and options available in the Sound of Vision (SoV) system, and were encouraged to use their preferred combination in each test.

The tests focused on evaluating the usability of the Sound of Vision system in real world outdoor scenarios. The main research questions addressed were:

1. *Are the visually impaired (VI) users able to perceive the environment (perception)? Are they able to identify obstacles and specific objects (negative obstacles, hanging obstacles, signs, walls) that define the added value of SoV compared to using the white cane? Is the system usable in real life environments and under real life circumstances (outside laboratory setups)?*
2. *Are the VI users able to use the information from the SoV device to guide their interaction with the environment (mobility)? Are they able to move around and avoid obstacles? Are they able to move around and identify targets (e.g., bus stop, corner of a building)? How is their mobility performance with the SoV system compared to traditional assistive devices (i.e., white cane)?*

The results of the evaluations were analyzed using a case study approach, which is more appropriate for understanding the strengths and weaknesses of the system, correlated to the individual user expectations and needs. This allows us to better explain the highly probable individual variance and inconsistencies in user performance and feedback given the small size of the sample involved in the tests vs. the high diversity of the visually impaired community. Further, an analysis of averaged results of all participants was also performed to overcome individual differences. However, statistical significance tests or extensive comparisons between demographic categories were not the purpose of the presented study. An emphasis was laid on acquiring user feedback through interviews and questionnaires.

While several research groups were involved in the multiple experimental and evaluation phases of the Sound of Vision project, each performed a specific study with the system in each phase [25–30]. Overall, almost 50 blind persons were involved in these evaluations throughout the project. However, the replication of the same experiments in real-life environments is difficult and prone to significant variations. In contrast, the experiments involving virtual environments and laboratory settings, were replicated at the premises of four partners in three countries. Furthermore, we do not provide a comparison between the results of the present study and those obtained in the previous usability evaluations with the system. This is mainly justified by the different versions of the SoV prototype used in the evaluations as several design–implementation–evaluation loops were employed during the project. Each loop was followed by improvements on the design and functionality of the system. Moreover, each evaluation phase had a different purpose, specific to the corresponding technological readiness level of the system.

The main contributions of this paper consist in providing a procedure for usability assessment of sensory substitution devices for the visually impaired in complex real-life environments as well as the results of applying it for the evaluation of the Sound of Vision SSD. Besides their intrinsic value for the validation of the multi-sensory feedback employed

in the Sound of Vision system, these results also provide further valuable insights for the development of any sensory substitution device for the visually impaired:

- Evaluation of environment perception and understanding based on audio and haptic feedback, in real life usage conditions;
- Evaluation of the usability of multi-sensory feedback of SSDs for mobility tasks in real life environments;
- Importance of training and recommendations for improved protocols and instruments for training with SSDs;
- Evaluation of the interplay between SSDs and traditional assistive instruments (white cane);
- Recommendations for the development of multi-sensory feedback systems for the visually impaired.

To the best of our knowledge, this is the first elaborate study on the usability of a sensory substitution device for the visually impaired in real-life conditions, outdoors.

## 3. Materials and Methods

### 3.1. The Sound of Vision System

The Sound of Vision system (SoV) is a wearable sensory substitution device (SSD) for visually impaired persons. The SSD aims to help them to understand the surrounding environment (perception) and to improve their mobility in unknown, indoor and outdoor environments (navigation). The SoV prototype used in the present study integrates custom and complex software and hardware solutions that enable real-time operation of the device. The system works by permanently acquiring environmental information through a fusion of cameras and sensors, extracting essential features and providing real-time feedback to the user by conveying the information through audio signals and haptics (vibrations)—Figure 1. Moreover, the system is designed to work in both indoor and outdoor environments and irrespective of the illumination conditions. A presentation of the SoV system can be accessed at https://youtu.be/6QRiwykp_bM (accessed on 5 July 2021).

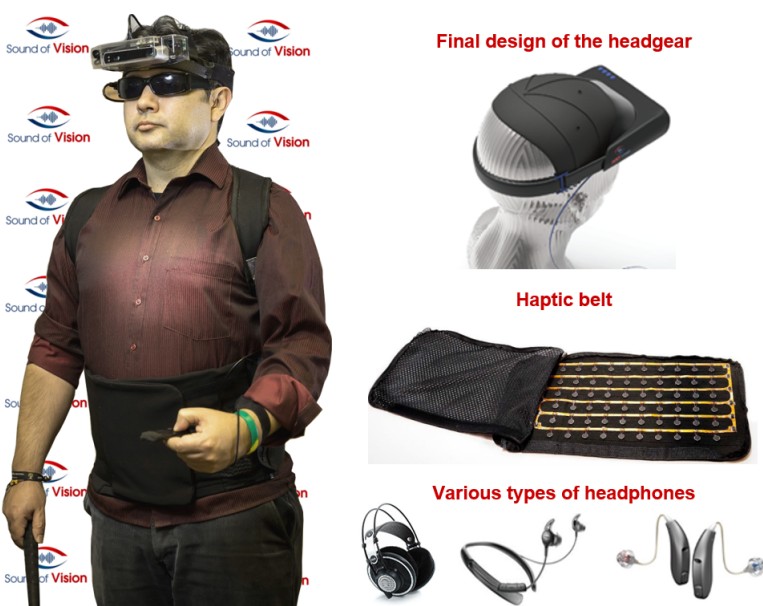

**Figure 1.** The Sound of Vision system. (**Left**) The prototype used in the usability evaluations. (**Right**) Various parts of the system.

Environment sensing and reconstruction is performed based on data acquired with different imaging and inertial sensors [31]. A structured light camera provides depth information in indoor environments and in low light or in the dark. In outdoor environments with normal and bright lighting conditions, the depth information is provided by a stereo vision system. This combination of sensors ensures the environment sensing in any

conditions. The 3D reconstruction and recognition of the various elements of interest in the environment is obtained with specific processing pipelines, tailored for the particular input used [23].

The 3D reconstruction pipeline used in the performed tests is based on stereo vision and is specifically tailored for outdoor environments. The details of the technical solution and the reconstruction algorithms for outdoor environments are presented in [24]. In this configuration, the system is able to identify various types of elements of interest and their properties (width, height and position with respect to the camera): generic obstacles, walls, negative obstacles (e.g., holes in the ground, stairs down), signs and texts.

Sound of Vision offers to the user several ways of perceiving the surrounding scene: 2 full scene encoders, plus tools useful for specific situations, and danger rendering. Changing between these and adjusting their audio/haptic options and volumes is easy to perform in real-time, using a remote control.

The full scene encoders are independent modes of encoding the information about all the objects in the scene (i.e., segmented and identified by the 3D module): iterative (renders the objects in a loop—one by one, in increasing order of distance from user, similar to an expanding sphere); continuous (renders all the objects simultaneously). In both cases, the rendering of each individual object has both audio and haptic outputs, which are carefully synchronized. Furthermore, both of the full scene encoders provide several options regarding the way that individual objects are rendered. For example, in the iterative mode, for audio, the user can choose between stimuli of two types: bar impact sounds or bubble sounds, while for haptics, the user can choose between the projection of shapes on the belt, or just of the closest points of the objects in the scene. The properties of the sound stimuli (e.g., pitch, duration, amount of oscillation, etc.) generated for each object intuitively encode its width, height, distance to the user and elevation from the ground [25]. For example, when encoded with impact sounds, the closer an obstacle is to the user, the louder its sound will be. The wider the obstacle, the deeper (lower) its sound will be.

The tools are designed in order to help the user in certain situations by providing simpler information; e.g., by reducing the number of objects in the scene that are encoded and rendered. The 'flashlight' tool provides a very simple encoding of the distance from camera to the first object touched by an imaginary line going straight out from the camera. It is helpful to carefully explore a scene and allows accurate perception of distances to objects' surfaces, and especially of their margins. The 'Best free space' tool gives the user a simple indication of the open space where he/she can navigate. Other tools offer the functionality of texts detection and reading, signs detection and encoding, and TTS scene description. Another important feedback provided to the user is represented by the notification for Dangers'. The system renders to the user, through acute, hard-to-miss stimuli, the potentially dangerous elements in the scene—i.e., head-level obstacles or holes on the ground that are located on collision course, in a specified range.

*3.2. Virtual Training and Testing Environment (VTE)*

Before testing the SoV final prototype in real-world scenarios, every user participated in a short training and testing session using the Virtual Training and Testing Environment (https://youtu.be/hBay25-KN10, accessed on 5 July 2021) [26,28]. The VTE offers a series of 3D scenes meant to train, evaluate and improve user skills and their familiarity with the SoV system. The VTE integrates three operating modes: learning—tasks are presented individually and the user can switch between various tasks, audio and haptic models, and through this mode the user can learn the audio/haptic feedback associated with the task; practice—the user pre-tests the learned information and receives feedback (correct/wrong) as well as additional details if required; and testing—the users test their ability to correctly answer the task, no feedback involved.

The main goals of the usability evaluation included in the present study concern the ability of visually impaired persons to use the system in the "wild", in outdoor environments. These represent the most complex environments for the SoV sensory substitution device and its users. Thus, prior to performing any training or testing in outdoor envi-

ronments, the users required to have a minimal experience with the system in the Virtual Testing and Testing Environment and indoor environments:

- The already trained users participated in one VTE and indoor training session, where they became familiar with the latest updates on the SoV system (changes or additions to the encodings, using the remote control, etc.).
- The new participants went through full training and testing in VTE (Single attributes, Frontal Pickups, Passing between, Treasure hunt—Boxes) and indoor (Frontal Pickups, Passing between, Treasure hunt—Boxes).

During the VTE and indoor training session(s), the users were presented with all the audio and haptic encodings available in the SoV system. After that, they were able to select the models according to their preferences and even switch between them during the training and testing sessions. This feature is available using the remote control by the visually impaired or by the trainer. The outdoor sessions contained both training and testing exercises. The goal of training was to make the users familiar with using the SoV system to identify real-world obstacles (both generic and special objects), both in ego-static scenarios and mobility scenarios. The training and testing scenarios contained both static and dynamic obstacles in natural outdoor scenes (in the parking lot, on the sidewalk, etc.) in usual environmental noise conditions. Each scenario started with training ego-static perception of the presented scene and continued with training the use of the system while moving in that scene.

Each outdoor session started with training in predefined scenarios and ended with a set of tests related to the scenarios trained in that session. For each scenario, the user trained the perception of the environment (i.e., ego-static), followed by training the mobility in that specific environment (i.e., ego-dynamic). For most of the scenarios, the egostatic training started with an environment containing only static objects, followed by adding dynamic objects, too. The dynamic obstacles were at first represented by moving persons (SoV team members). Uncontrolled environments were gradually added to the training and testing where dynamic obstacles were represented by any person, bicycle, or car passing by. The testing scenarios were different from the trained ones to minimize the effect of learning the environment. At most, it was acceptable to use a trained setup, but starting with a different position and orientation for the user. Figure 2 illustrates a map of the University campus and downtown Iasi where the locations of the training and testing environments are marked.

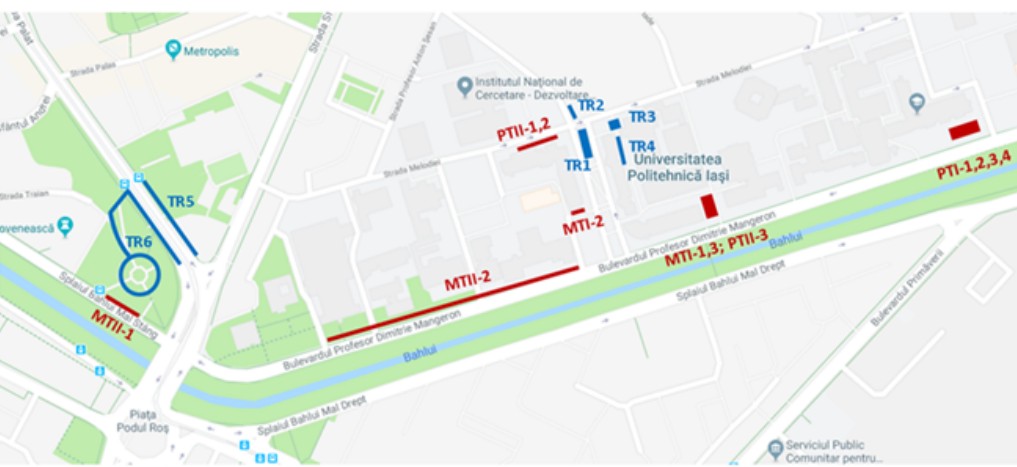

**Figure 2.** Geographical localization of the environments used for training and testing.

The outdoor training time was minimal, it had small variations between users, depending on their skills with the system. There was a number of 4 sessions of 2 h planned for training (TR) and testing with each user (Table 1). However, due to weather conditions (temperatures approaching 0 °C making some of the testing equipment work inappropri-

ately) and users'availability, some sessions were shorter than 2 h, and thus some users were invited for one extra session.

**Table 1.** Overview of the training and testing sessions in outdoor environments (TR—training session, PT—perception testing session, MT—mobility testing session).

| Session | Training | Testing |
|:---:|:---:|:---:|
| 1 | Training in the parking lot—generic objects, static and dynamic (environment TR1); training with elevated objects—tree branches (environment TR 4) | Testing PT I |
| 2 | Training with walls (environment TR 4); training with negative obstacles—holes in the ground (environments TR2 and TR3) | Testing PT II |
| 3 | Training in the parking lot—generic objects, static and dynamic(TR1), on the sidewalk, finding the bus stop (environments TR6) | Testing MT I |
| 4 | Training mobility on the sidewalk, finding the bus stop (environment TR5) | Testing MT II |

### 3.3. Data Collected During the Experiments

For each test, the following data was collected: use of encodings, electrodermal activity (Galvanic Skin Response, Heart Rate)—only during mobility tests, video footage, and user feedback. The electrodermal activity data was collected using a device (GRS + Shimmer) that is mounted on the fingers and hand. This setup incapacitates the use of the remote control with the respective hand. The other hand is used to hold the white cane in the corresponding tests. Thus, it was decided that the users would not change the selected encodings during a test. However, they are able to choose their favorite combination before each test, after the task is explained to them. The video footage was processed offline to extract the completion time and accuracy metrics for each test. After each test, the users responded to the task related questions. The users also responded to the system general questions, once after the finalization of the Perception Tests (PT) and once after the finalization of the Mobility Tests (MT).

### 3.4. Testing Equipment

The equipment used in the testing sessions was composed of the following devices:

- SoV system (using as main input the video feed from the stereo camera);
- GRS + Shimmer for electrodermal activity recording;
- SoV Test Utility for automatic recording of encodings usage and time (application running on the SoV device);
- Tablet for remote connection to the SoV device (used for inspection of system status, for starting/stopping the recording of the test session using the SoV Test Utility, for starting/stopping the recording of electrodermal activity data);
- Wi-Fi bundle to ensure connection between the SoV device and the tablet.

### 3.5. Study Design

The evaluation tests are divided in two categories: Perception Tests (PT)—evaluation of the SoV prototype and assessment of its usability compared to the white cane for perception; Mobility Tests (MT)—evaluation of SoV usability for mobility.

Some mobility testing scenarios were performed by the users in three conditions: (1) with the white cane only, (2) with SoV and white cane, and (3) with SoV only. This allowed a comparison between the performance in the three use cases. To this end, some testing scenarios (i.e., particular environments) were carefully selected such that they contained the same structure of the environment in all cases. Still, outdoor environments are highly dynamic in change. Thus, testing in such scenarios can also pose a high uncontrollable variance between different tests and users, even if following the same course in the same environment. To account for the variance, the analysis of averaged results across users was performed by weighting the results based on the difficulty of the course. The difficulty level was assigned based on the total number of obstacles. In order to collect meaningful

data, some testing scenarios were "fabricated". However, they still contained real-world objects and resembled for as much as possible situations that users can encounter in the "wild". Operating in such semi-controlled environments also helped ensure the safety of the visually impaired participants. For safety reasons, during all training and testing sessions, the participants were closely assisted by at least one sighted test assistant. The role of the test assistant was to take care of the safety of the test-taker and stop him/her before running into any dangerous situations.

The perception tests (Table 2) were aimed at evaluating how the visually impaired participants perceive the environment in ego-static scenarios (i.e., the user is standing in a fixed position). They were set in realistic outdoor environments, with various noise conditions, containing static and dynamic elements (cars, poles, trees, people, buildings), as well as generic and special (walls, holes in the ground, signs) types of objects that can be identified and signaled by the SoV device. For safety reasons, the training and testing sessions with negative obstacles (holes in the ground) were performed without the user moving in the scene. The identification of negative obstacles was only evaluated in the tests in the perception category and not in the mobility one. The distance and direction for dynamic objects were not evaluated (a dynamic object is an object that moves in the scene, changing its direction and/or distance to the user).

**Table 2.** Overview of the outdoor tests designed for evaluation of environment perception with the SoV device (PT—perception test).

| Setting | Test ID | Scenario Details | Tasks | Performance Metrics |
|---|---|---|---|---|
| **PT I**—complex scenes with generic objects (static, dynamic, hanging) | PT I-1 | Generic static objects (car, person, bush, tree, pole) | | |
| | PT I-2 | Generic static and dynamic objects | The user is asked to identify: | |
| | PT I-3 | Generic static and elevated objects | How many objects are present in the scene | |
| | PT I-4 | Generic static, dynamic and elevated objects | | Accuracy |
| **PT II**—complex scenes with generic objects and special objects (static, dynamic, negative obstacles, walls) | PT II-1 | Generic static and special objects (negative obstacles: hole in the ground) | Type of objects | Completion time (static, dynamic, special) |
| | PT II-2 | Generic static, dynamic and special objects (negative obstacles: hole in the ground) | Localization of objects (distance, direction) | |
| | PT II-3 | Generic static and special objects (wall) | Elevation of objects | |

The mobility tests (ego-dynamic scenarios) were aimed at evaluating how well the users can guide their interaction with the environments based on the information received from the SoV device. Two types of scenarios were considered: semi-controlled environments (Table 3 MT I) and uncontrolled environments (Table 3 MT II). The users were asked to walk on predefined routes fulfilling specific tasks. By semicontrolled environments, we denote natural areas, usually containing short testing routes (15–30 m long), with no or light traffic, for which the testing team could control the structure of the scene. This ensured the presentation of the same scene and tasks to all participants. Specific static and/or dynamic obstacles (i.e., people) were purposely and systematically introduced in some testing scenarios. By uncontrolled environments we understand public areas, with varying, uncontrollable traffic. This only allows for qualitative (as opposed to quantitative) performance evaluations and comparison of performance between users and modalities

(i.e., white cane, SoV + cane, SoV). Still, it better reflects the usability of the system in the targeted real-life environments.

**Table 3.** Overview of the tests designed for evaluation of mobility in outdoor environments with the SoV device (MT—mobility test).

| Setting | Test ID | Scenario Details | Tasks | Performance Metrics |
|---|---|---|---|---|
| **MT I**—semi-controlled environments (static, dynamic, hanging objects, walls) | MT I-1 | Walking by a wall (30 m path): static obstacles on the path | The user is asked to: Identify the wall; Walk along the wall at a certain maximum distance from it; | |
| | MT I-2 | Walking by a wall (15 m path): hanging obstacles on the path | Identify and avoid the obstacle(s) on the path; Identify the corner of the wall. | |
| | MT I-3 | Walking in a parking lot (25 m path): parked cars, dynamic obstacles on the path | The user is asked to: Walk along parked cars at a certain maximum distance from them; Identify the last parked car on the left/right of the course; Identify and avoid dynamic obstacles on the path. | Accuracy Time to completion Collisions |
| **MT II**—uncontrolled environments (static, dynamic, hanging objects, signs) | MT II-1 | Walking on the sidewalk (45 m path): static and dynamic obstacles on the path, bus stop | The user is asked to: Walk on the sidewalk; Identify and avoid static and dynamic obstacles; Identify and stop at the bus stop | |
| | MT II-2 | Walking on the sidewalk (250 m path): static and dynamic obstacles on the path | The user is asked to: Walk on the sidewalk; Identify and avoid static and dynamic obstacles; | |

### 3.6. Description of the Perception Experiments

Only two types of scenarios were selected for evaluation: complex scenes with generic objects (Table 2 PT I) and complex scenes with generic and special objects (Table 2 PT II). These are ego-static tests, where the visually impaired user is standing at a fixed point within the real-world environment and interprets the varying scenes presented to him/her. The user is only relying on the SoV system. An overview of the scenes selected for testing is presented in Table 2. Even though the user is not changing their position, he/she can look around in the environment. This is important, as head turning is a fundamental part of orientation for sighted people, with the role of expanding the visual field. The SoV training program encourages visually impaired people to use the device in the same way. Moreover, unlike the tests performed previously in Virtual Testing Environment [28] (VTE) and indoor real-world, the outdoor tests also included the presence of dynamic obstacles. The main goals of the perception tests were to evaluate whether the visually impaired participants:

1. Are able to perceive the environment using the SoV device from a stationary position.
2. Are able to identify obstacles and recognize specific objects in static environments.
3. Are able to identify both static and dynamic obstacles and recognize specific objects in real-life environments.

The first type of perception tests (PT I) included only generic objects. These objects were represented by people, cars and poles (Figure 3). The tests were performed in an order of gradually increasing difficulty, starting from a scene with static objects (PT I-1, Figure 3a), and then adding a dynamic object (PT I-2, Figure 3b). In the next step, the perception of elevated objects was tested, first in a static scene (PT I-3, Figure 3c), then in a scene with dynamic objects (PT I-4, Figure 3d). A piece of cardboard was held by one test

assistant at head level to represent the elevated object. The second type of perception tests (PT II) included generic and special objects. The special objects considered were walls and negative obstacles, i.e., a hole in the ground (Figure 4). The hole in the ground was represented by a sewer for which the cap was removed for the purpose of the tests.

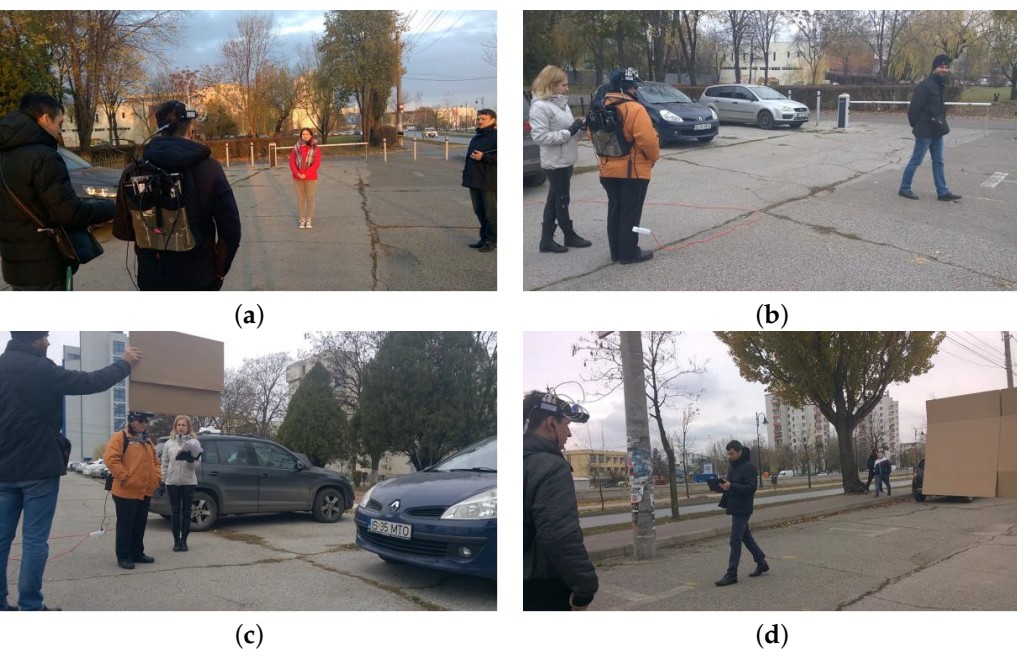

**Figure 3.** Examples of testing scene setups for perception tests with generic objects: (**a**) Scene with generic static objects (PT I-1); (**b**) scene with generic static and dynamic objects (PT I-2); (**c**) scene with generic static and elevated objects (PT I-3); (**d**) scene with generic static, dynamic and elevated objects (PT I-4).

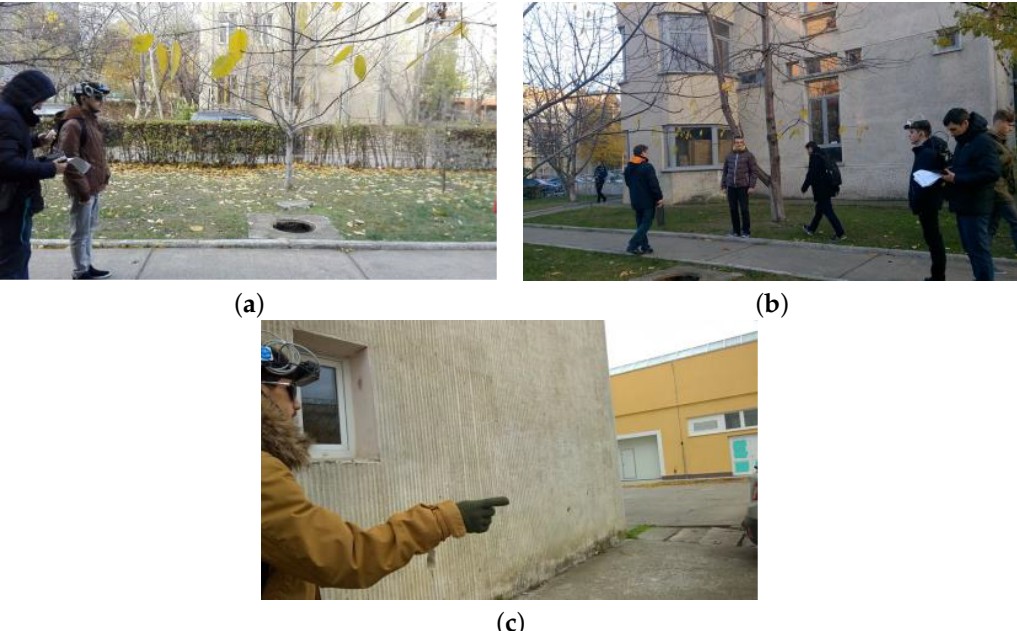

**Figure 4.** Examples of testing scene setups for perception tests with special objects: (**a**) scene with generic static objects and hole in the ground (PT II-1); (**b**) scene with generic static, dynamic objects and hole in the ground (PT II-2); (**c**) scene with generic static objects and wall (PT II-3).

The system was used with a fixed distance range of 5 m. The scenes contained 3 to 5 objects in this range, with varying distance and orientation to the user. Some tests were performed in the same physical location (e.g., parking lot). However, the user position

and orientation were changed to maximize the difference between consecutive test scene layouts. The system output was paused while preparing the testing setup or switching between consecutive testing scenarios. The system output was turned on only after the testing setup was prepared and the user was positioned and oriented accordingly.

Before starting the test, users are instructed:

- To stand still and analyze the presented scene by scanning with the head.
- To inform the testers immediately in case the representation stops or the audio sounds are distorted, etc. In that case, the users are asked to stop, and the testers pause the completion time while fixing the problem. In case it can be fixed directly, the user is asked to proceed with the scene, and the time measurement continues.
- For each tested scenario, the user is informed about the context and the tasks he/she is supposed to perform:

  *You will be presented with a scene including several objects at the same time, each with varying distance, direction, size and quantity. You will hear (feel) how the scene is represented with sounds (vibration patterns). Your task is to try to understand the scene, the relation between the different objects, trying to get an inner picture of its composition. You will be asked the following questions for each scene in random order, one at a time. As soon as one of these questions is asked, you should answer it verbally:*

  1. *How many objects do you perceive?*
  2. *Out of the perceived objects, how many are special objects (i.e., wall, hole in the ground)?*
  3. *If any special objects, specify their type.*
  4. *Out of the perceived objects, how many are dynamic?*
  5. *For each static object, indicate: the distance to the object, the direction of the object, the elevation of the object.*

- To choose their favorite encoding for audio and haptic, which they are not be able to change during the test.

*3.7. Description of the Mobility Experiments*

The mobility tests included two types of scenarios: semi-controlled environments (MT I) and uncontrolled environments (MT II). These are ego-dynamic tests, where the VI user is asked to walk on a predefined route fulfilling specific tasks. An overview of the scenes selected for testing is presented in Table 3.

The main goals of the mobility tests were to evaluate:

1. *Whether the visually impaired participants are able to use the information from the SoV device to guide their interaction with the environment.*
2. *Whether the visually impaired participants are able to move around and avoid obstacles using the SoV device.*
3. *The efficiency of navigation with SoV device compared to using only the cane.*

For two testing scenarios, i.e., MT I-1 and MT II-2, the course was performed in three conditions: (1) only white cane, (2) SoV + white cane, (3) only SoV. Scenarios MT I-3 and MT II-1 were performed in conditions (2) and (3). Using only the white cane cannot solve scenarios involving the detection of hanging objects or signs. Thus, in testing these scenarios, the condition of using only the white cane was omitted.

The first type of mobility tests (MT I) were performed in semi-controlled environments (Figure 5). The tests evaluated the usability of the system on short paths in natural identical set-ups for all users. For scenario MT I-3, a dynamic obstacle was introduced in the scene by having a SoV team member walk towards the user on their path. The second type of mobility tests (MT II) were also performed on the same courses for all users and for all modalities. However, the environment was not controlled in any way (Figure 6). The difficulty of the course with respect to the number of obstacles varied between different instances of the tests. To account for this variability when comparing the performance between users and between different modalities on the same course (white cane, SoV and white cane, SoV), the performance metrics (time and collisions) were adjusted based on

the number of obstacles present on the course (as described in the Performance Metrics section below).

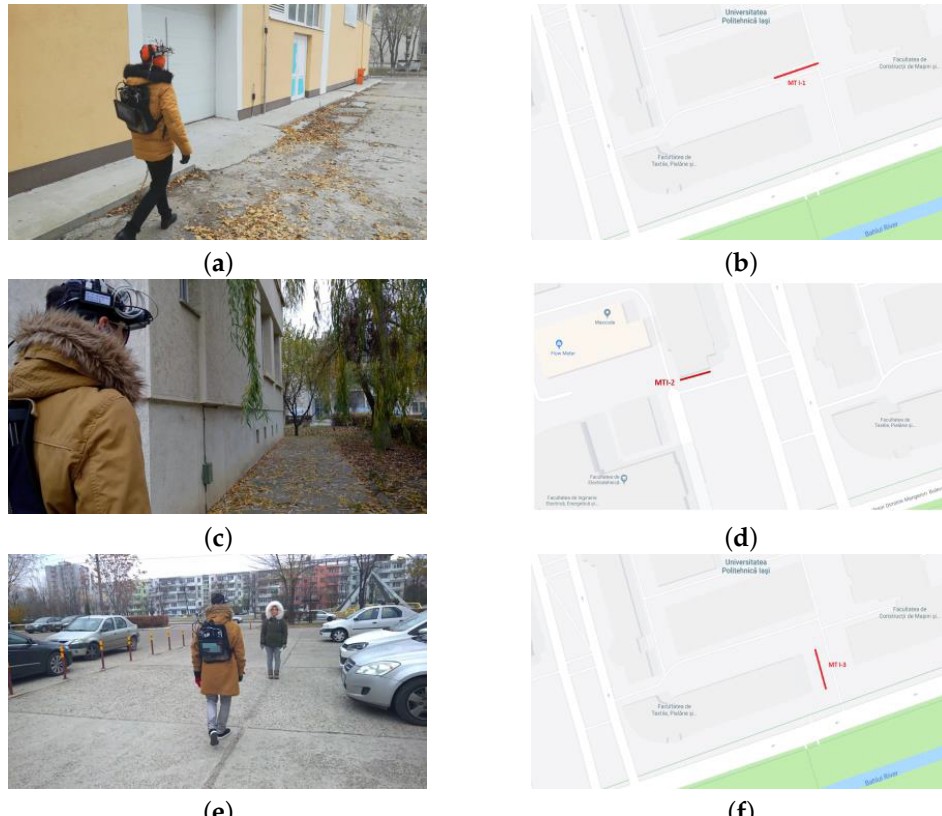

**Figure 5.** Examples of testing scene set ups for mobility tests in semi-controlled environments: (**a**) Walking by a wall (30 m path), static obstacles on the path (MT I-1); (**b**) Geolocation of MT I-1 testing scenario; (**c**) Walking by a wall (15 m path), hanging obstacles on the path (MT I-2); (**d**) Geolocation of MT I-2 testing scenario; (**e**) Walking in a parking lot (25 m path), parked cars, dynamic obstacles on the path (MT I-3); (**f**) Geolocation of MT I-3 testing scenario.

The system output was paused while preparing the testing setup or switching between consecutive testing scenarios. The system output was turned on only after the testing setup was prepared and the user was positioned and oriented accordingly.

Before starting the test, users are instructed:

- To avoid collisions with obstacles as well as with walls on any sides.
- To inform the testers immediately in case the representation stops or the audio sounds distorted, etc. In that case, the users are asked to stop, and the testers pause the time while fixing the problem. In case it can be fixed directly, the user is asked to proceed with the scene and the time measurement continues.
- To choose their favorite encoding for audio and haptic, which they are not able to change during the test.
- For each tested scenario, the user is informed about the context and the tasks he/she is supposed to perform:

    1. *MT I-1: You are standing in close vicinity of a building. Please identify its wall and walk along the wall at a comfortable distance to it, no more than 2m. There might be static and/or dynamic obstacles on the path that you should avoid. Please indicate verbally when you have reached the corner of the building.*
    2. *MT I-2: Same as for MT I-1.*
    3. *MT I-3: You are standing in a parking lot. There are parked cars to your left/right. Walk along them until you identify the last parked car on the left/right of the course. There*

might also be other static and/or dynamic obstacles on the path that you should avoid. Indicate verbally when you have reached the last car.

4.   *MT II-1: You are standing on the sidewalk. Walk on the sidewalk in front of you until you reach the bus stop. There might be static and/or dynamic obstacles on the path that you should avoid. Indicate verbally when you have reached the bus stop and point to the direction of the bus stop sign. You will be closely assisted by two test assistants who will stop you in case of any unsafe situation.*

5.   *MT II-2: You are standing on the sidewalk. Walk on the sidewalk in front of you until you are told to stop (about 250 m). There will be static and/or dynamic obstacles on the path that you should avoid. There are two small side streets that you will cross, which are not equipped with traffic lights or pedestrian crossings. Report if you think it is safe to cross, otherwise stop. You will be closely assisted by two test assistants who will stop you in case of any unsafe situation.*

The users were NOT informed about the scene layout, meaning there were no instructions on the number, location and size of objects or about the length of the course they are supposed to follow.

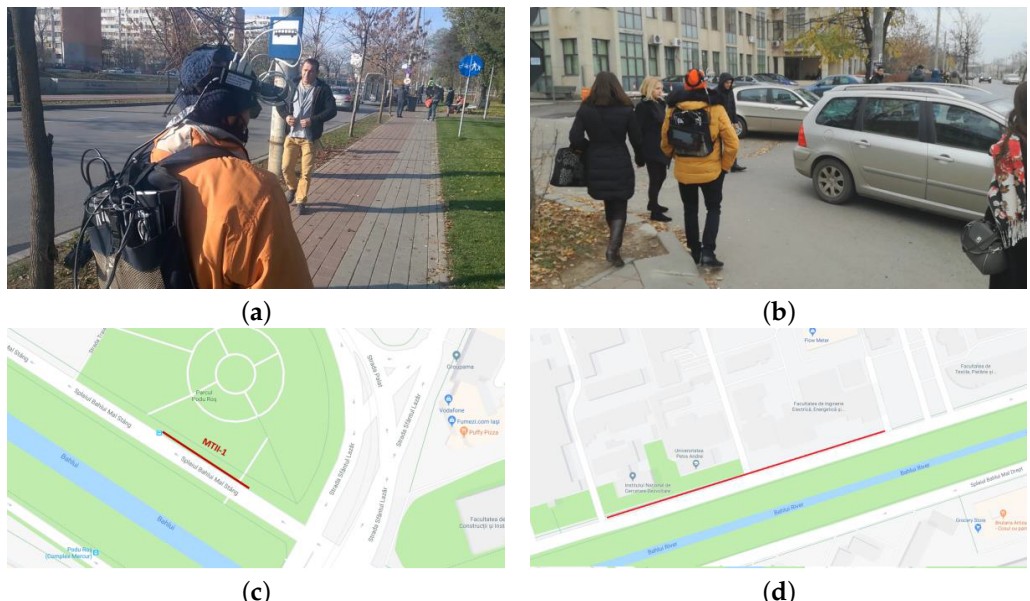

**Figure 6.** Examples of testing scene set ups for mobility tests in uncontrolled environments: (**a**) Walking on the sidewalk (45 m path), static and dynamic obstacles on the path, bus stop (MT II-1); (**b**) walking on the sidewalk (250 m path), static and dynamic obstacles on the path (MT II-2); (**c**) geolocation of MT II-1 testing scenario; (**d**) geolocation of MT II-2 testing scenario.

*3.8. Performance Metrics*

The following metrics were used to assess the performance for the perception tests:

- The accuracy is computed based on correct or incorrect answers given by the user to each of the addressed questions in each tested scenario.
- Completion time is computed as the time between the moment when the SoV system is turned on at the beginning of the test (this is also when the user starts perceiving the environment) and when the answer to the last question is provided by the user. That is, the completion time is measured per test and not per task.

A perception test is considered as completed with success if the average accuracy over all the tasks (questions) in the test is greater than 85%.

The metrics used to assess the performance for the mobility tests are:

- The accuracy for a task is reflected by identification of the elements of interest (e.g., wall, corner of the building, hanging object, bus stop), and/or leaving the

testing area (i.e., distancing from the indicated shoreline, leaving the sidewalk by walking to the street).

- The number of collisions, where only major collisions with obstacles are considered. By major collisions, we mean contacts with objects the participant was completely surprised about, while minor collisions are considered when the user brushes the obstacles.
- The number of cane hits was accounted for in some scenes, considering only the hits on obstacles.
- Completion time is measured between the moment when the SoV system is turned on at the beginning of the test (this is also when the user starts perceiving the environment) and when the task is finalized with success.

A mobility test is considered as completed with success if the elements of interest have been identified and the user has not left the testing area, irrespective of the number of collisions or completion time. Due to the high variance of the number of obstacles on the MT II-2 course between the tests performed with different users and between tests with different modalities for each user, a supplemental measure was introduced: *Adjusted Completion Time*. This measure allows for comparing between different instances of the same test, by weighting the actual measured time taken to complete the course with the number of extra obstacles found on the course. We consider that the course contains a fixed number of obstacles in all instances (represented by trees, poles, benches, bushes), besides which, a number of extra obstacles, static or dynamic, were present on the course (people, parked cars, dynamic cars, bicycles). Thus, we define the *Adjusted Completion Time (ACT)* for scenario MT II-2, to be:

$$ACT_{MTII-2} = CT \times \frac{O_a}{O_{crt}}, \tag{1}$$

where $CT$ is the measured completion time, $O_a$ is the average number of extra obstacles, over all test instances of all users in mobility scenario MT II-2, and $O_{crt}$ is the number of extra obstacles in the current test instance. Thus, if the number of extra obstacles in one test is higher than the average number of extra obstacles over all MT II-2 tests, the *ACT* value will be lower than the measured completion time (*CT*).

*3.9. Ethical Aspects*

All the investigations of the present study that involved tests with human subjects were carried out following the rules of the Declaration of Helsinki of 1975 (https://www.wma.net/what-we-do/medical-ethics/declaration-of-helsinki/, accessed on 5 July 2021), revised in 2013. According to point 23 of this declaration, an approval from the Research Ethics Committee of the "Gheorghe Asachi" Technical University of Iasi, Romania, was obtained before undertaking the research: research ethics assent no. 13582/05.07.2016 for the activities of the project no. 643636 (H2020), Sound of Vision—Natural Sense of Vision Through Acoustics and Haptics. Before starting any testing (e.g., questionnaire or "first hands on" period), the participants were informed about the project aims, the general function of the device and the aim of the tests. Further, the participants were informed about the test methods, their tasks and about possible risks involved in the testing but also the project teams efforts to minimize them. Additionally, the participants could agree on being filmed for the project purposes. Afterwards, the participants confirmed that they were informed and agreed on participating by signing the informed consent. According to the ethical approval requirements, that restrict reporting of the individual data, in the following, all users will be referred to as male, independently of their gender. Further, the age of the users will not be reported individually but they will be referred to as young (20–30), middle-aged (31–39) and older users (40–50).

## 4. Results of the Usability Experiments

Four visually impaired users, aged between 20 and 42, one female and three male, participated in training and testing sessions which took place at the Technical University

of Iasi. An overview of relevant information regarding the user profiles is presented in Table 4. Three of the testers were already familiar with the SoV system, since they were also invited to the training and testing sessions with an early SoV prototype. However, because improvements have been made (hardware, software, audio and haptic models) since the last sessions of training/testing, these users undergone one training session in VTE and indoor in order to refresh their memory and to present them the new features of the system. Furthermore, a new user, without any previous experience with the SoV system participated in the training/testing sessions. The newcomer followed the procedure of training and testing in VTE and indoor before outdoor sessions. He had to familiarize with the system and to learn its operating modes (audio and haptic). The users belonged to categories 4 and 5 of visual impairment as defined by the World Health Organization (WHO). Two users belonged to category 4, a category of blindness, meaning that visual acuity is less than 10% (FC at 1 m) and equal to or better than light perception. Two users belonged to category 5, total blindness, meaning no light perception. None of the 4 users use echolocation to guide their interaction with the environment. Users of category 4 were NOT blindfolded during the training and testing sessions. The main purpose of the tests was to evaluate the usability of the system in the "wild". Thus, blindfolding the users would be in contradiction with the realistic way in which they would use the system.

Due to light conditions, weather and also being restricted by users' time to participate both in training and testing in outdoor scenarios, the results detailed below are obtained with a minimum training time. With proper training, we assume that users will be able to improve and will obtain better or even excellent results. Thus, it is important to keep in mind that the results listed are based on minimal training in outdoor scenarios.

Average accuracy and completion times in the perception tasks are presented in Figure 7a,b. The most time consuming test was PT I-2 (first complex scene in which dynamic objects were introduced), while the smallest accuracy was recorded for the PT I-3 scene (first complex scene in which elevated objects were introduced). However, perceiving elevation of objects was not the most difficult task. Moreover, the perception of almost all object properties improved from PT I to PT II thanks to the additional training time with the system between the testing sessions. The only exception is represented by the dynamicity of objects. Due to the powerful and distracting sound made by the negative obstacles, 3 out of 4 users were not able to correctly identify if the scene contains a dynamic object or not, when holes in the ground were present. However, users ranked above 80% accuracy regarding special objects: they obtained 100% accuracy in counting them, 83% accuracy in indicating the correct distance and 91.6% accuracy in indicating the correct direction.

The perception tests conducted in ego-static scenarios helped us to conclude that the participants using the SoV device are able to perceive the environment even if they are not moving, and all users obtaining an average accuracy greater or equal to 85% in both types of scenes. The overall average accuracy in the perception tests was 89.3%.

Thanks to the improvements in audio models (danger mode and to different encodings for special objects) users correctly identified obstacles (possible dangers) and special objects (hole, wall) in static environments. As a result, all of them correctly identified the special objects (100% accuracy) and three out of four users correctly pointed to the direction and approximated the distance to the special objects. As we have mentioned before, in complex scenes, where both negative obstacles and dynamic obstacles are present, participants had difficulties in identifying the presence of a dynamic object in the scene (mostly because their attention is directed to the negative obstacle), but the accuracy of identifying and localizing static objects increased even with a small amount of training. Moreover, the perception of dynamic objects was the least trained aspect in the VTE sessions, even no training at all was performed by User 4.

**Table 4.** Description of the SoV users.

| | User 1 | User 2 | User 3 | User 4 |
|---|---|---|---|---|
| **age: young, middle-age, old** | middle-age | old | young | young |
| **visual impairment category (according to WHO)** | 4 | 5 | 5 | 4 |
| **travelling in unknown environments** | accompanied by sighted person | accompanied by sighted person | accompanied by sighted person | accompanied by sighted person |
| **white cane user** | uses the white cane but is not an experienced user | uses the white cane but is not an experienced user | no | no |
| **level of experience** | training and testing beginning with the first prototype | training and testing beginning with the first prototype | training and testing with the prototype before the final one | no previous training and testing |
| **preferred audio model** | the expanding sphere model with the impact sounds | the expanding sphere model with the bubble sounds and the flashlight audio encoding | the expanding sphere model with the bubble sounds, the bubblestream and flashlight | the expanding sphere model with the impact sounds |
| **preferred haptic model** | closest point | closest point | closest point | closest point |
| **others** | likes to sing, passionate about smartphones and IT technologies | active member of the local blind community, participates in different competitions and social activities dedicated to visually impaired | studies foreign languages | loves music, plays 7 instruments (e.g., flute, pan, oboe), studies pan flute at the Arts University, participates in various cultural activities |

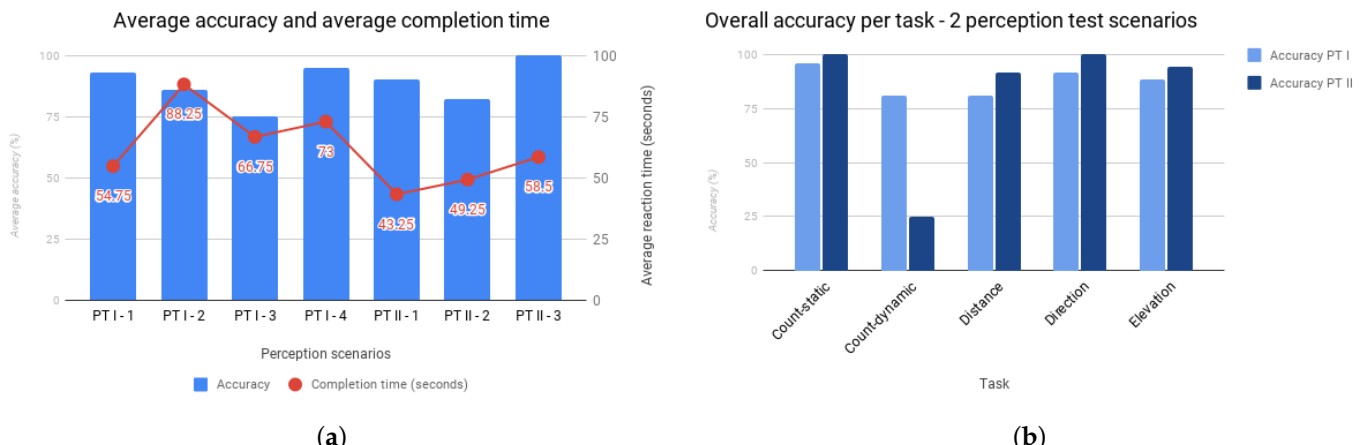

<div align="center">(<b>a</b>)　　　　　　　　　　　　　　　　　　　　　　　　　　　　　(<b>b</b>)</div>

**Figure 7.** (**a**) Average accuracy and completion time over all users in perception tests (**b**) Average per task accuracy over all users in both perception scenarios.

Another important aspect addressed in the evaluation was the comparison between SoV and the white cane in mobility scenarios. To this end, scenarios MT I (walking by a wall) and MTII-2 (walking on the sidewalk) were employed with three modalities: using the SoV device together with the white cane, only using the white cane, only using the SoV device. As expected, the average time for completion was the lowest when using the white cane in both scenarios (Figures 8 and 9). With the SoV system, users can walk at a reasonable distance from the wall in MT I, without the need to hit it with the cane

(average of 23 cane hits when using the white cane only, compared to 3.5 hits when using the white cane and SoV) and could easily detect the corners of the buildings. Users with no experience of either of the two devices tend to walk slower when using both modalities than only with either of them, as the amount of information coming from both devices can be overwhelming. This is also confirmed by the results of the MT II-2 test: the average time (adjusted based on the average number of additional obstacles on the course in each modality) to complete the course in the conditions with a single assistive device (cane only and SoV only) was very similar, but much less than in the condition with both devices.

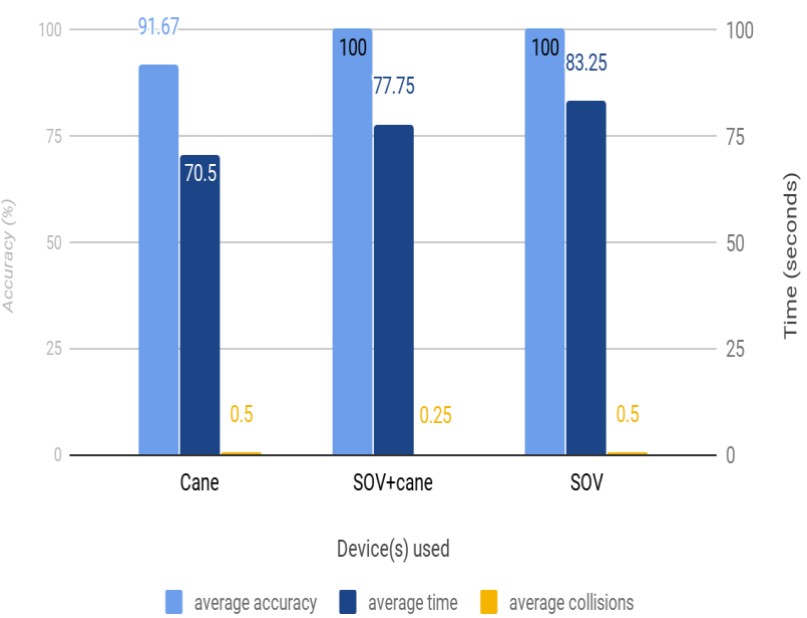

**Figure 8.** Average accuracy, collisions and completion time over all users in mobility scenario MT I-1 (walking by a wall) when using the SoV system alone, SoV and white cane, white cane alone.

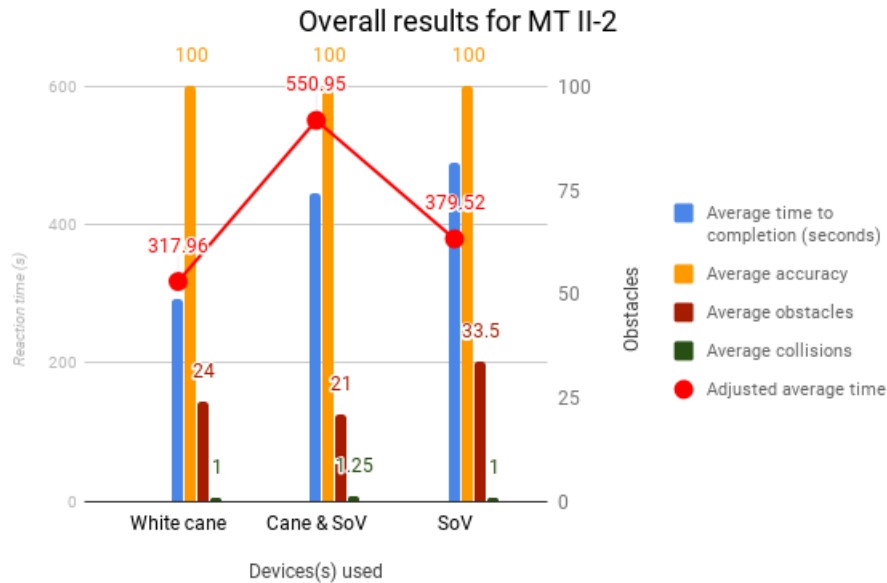

**Figure 9.** Average results over all users for the MT II-2 scenario (walking on the sidewalk) with the 3 modalities: SoV, white cane, SoV and white cane.

An overall measure for expressing the completion rate of the tasks with a system is the *effectiveness*. It can be used to evaluate the current version of the SoV device, being a measure that embeds accuracy and task completion. Considering the following notations *N*-the number of the scenarios, *R*-the number of users, $n_{ij}$ the result of coming through scenario *i* by respondent *j*, if the user successfully completes the task, then $n_{ij} = 1$, if not, then $n_{ij} = 0$, $t_{ij}$ the time spent by user *j* to complete task *i*. If the task is not successfully completed, then time is measured until the moment the user quits the task; the overall product effectiveness *E* can be computed using:

$$E = \frac{\sum_{j=1}^{R} \sum_{i=1}^{N} n_{ij}}{R \times N} \times 100\%, \tag{2}$$

its statistic error being:

$$\sigma = \sqrt{\frac{E \times (100 - E)}{R}} \tag{3}$$

The data used for computing the effectiveness of the SoV system was obtained by merging all tests (perception and mobility). A number of 65 scenarios for all four users were considered. If a task *i* is performed by user *j* with accuracy greater than 85%, then $n_{ij} = 1$. Otherwise, $n_{ij} = 0$. Based on this data, the effectiveness of the SoV system is $E = 88.85\%$, with a statistic error $\sigma = 15.74\%$.

Taking into account that for successful task completion a high accuracy threshold of 85% was considered, the value of the obtained effectiveness underlines a very good overall performance of the SoV system.

An exploration of the individual performance of each of the four users is provided as Supplementary Material for the paper.

## 5. Discussion

To summarize, the initial questions we posed were all addressed with the performed tests. In the following, we provide a discussion of the combined results with respect to each aspect that was evaluated.

*Are the visually impaired (VI) users able to perceive the environment (perception)? Are they able to identify obstacles and specific objects (negative obstacles, hanging obstacles, signs, walls) which define the added value of SoV compared to using the white cane? Is the system usable in real life environments and under real life circumstances (outside laboratory setups)?*

Identifying the objects in the environment and their individual properties (position, size, elevation) is important for both perception and mobility. Perception of the environment with the SoV system was therefore evaluated in both ego-static and ego-dynamic scenarios. Evaluation of perception was specifically addressed in ego-static scenarios in the virtual training environment and the tests performed in real-world scenarios. The perception tests revealed impressive accuracy scores in counting the objects in real-world complex scenes (97.5%). The participants could also identify their distance well (86.45% accuracy) and direction (95.8% accuracy). Detecting the elevation of objects is important for avoiding head-height obstacles. As users noted, it is sufficient to be aware of the presence of the obstacle, not necessarily its exact elevation. Still, understanding this property with the help of the SoV device appeared to be easy, given the 91% accuracy obtained in real-world outdoor tests.

Identification of the presence of special objects in the scene was performed with very good results. Localizing walls proved to be easy to perform with the SoV system (100% accuracy obtained in the tests). This was also the case for identifying holes in the ground, represented by missing sewer caps in real-world outdoor tests (100% accuracy in identifying their presence in the scene, 87.5% accuracy in determining their direction and 75% for their distance). Another added value of the SoV device compared to using the white cane lies in the identification of signs. This aspect was tested as part of a mobility scenario involving walking on the sidewalk and identifying a bus stop. Remarkably, this

task was completed with a 100% success rate and was considered very easy to perform by all the participants.

The perception of dynamic obstacles was also evaluated. It seems that identifying the dynamicity of objects (i.e., whether an object is moving or not) is more difficult to perform when special objects (e.g., holes in the ground) are also present in the scene. As the participants themselves explained, the negative obstacles are represented with a powerful sound that draws most of their attention. Aspects that require more concentration can therefore be misinterpreted. Still, it is important to note here that the perception of dynamic objects was the least trained aspect, so there is room for improvement.

The tests performed outside laboratory setups revealed that the individual properties of objects in complex scenes can still be perceived, even in natural outdoor scenes in the presence of high environmental noise. The participants in the tests obtained an overall 89.3% success rate in identifying object properties in such scenarios. The easiest to understand were the number and direction of objects, while the most difficult aspect was whether an object was moving or not (dynamicity).

*Are the VI users able to use the information from the SoV device to guide their interaction with the environment (mobility)? Are they able to move around and avoid obstacles? Are they able to move around and identify targets (e.g., bus stop, corner of a building)? How is their mobility performance with the SoV system compared to traditional assistive devices (i.e., white cane)?*

Mobility in outdoor environments was evaluated in the tests performed. The main aim was to assess the system usability in real-life environments and also its added value compared to the white cane. The results of these tests show that the SoV system offers the clear advantage of informing users about the presence of objects which could not be otherwise detected with the cane (head level objects, signs) or that could be missed by it (holes in the ground). Moreover, the system provides a good solution for detecting walls, which are frequently used by VI people as a shoreline during navigation. With the SoV system, users can walk at a reasonable distance from the wall, without the need to hit it with the cane and could easily detect the corners of the buildings. Head level obstacles were identified and avoided with 100% success rate in the tests. The primary reason for failure was when the participants walked too fast while "looking" down, so that the head-height obstacles (i.e., tree branches) were out of the camera's field of view until they were very close. This result also emphasizes the need to train the VI users to hold their head in positions similar to those of the sighted and scan the environment with the cameras like the sighted do with their eyes.

It is important to note that, while most of the participants had some level of experience with the system in VTEs and indoors, their outdoor training time was minimal (average of 2 h).

The SoV system offers a rich perception of the environment, which is not by far available with the white cane. White cane users have no to very little information about the environment especially in static scenes (where they do not move). They can perceive only the objects in their immediate proximity, as far as they can scan the scene with the white cane, no information is available further than the length of the white cane. On the other hand, the SoV system offers the possibility to acquire information about surrounding objects further away from the user (5–10 m), so one can have a better, more complex and early understanding about the environment. Furthermore, even when users cannot successfully identify if an object is hanging, they are still able to perceive its presence, while with white cane it is almost impossible to detect a hanging object without bumping into it.

With minimal training on the SoV prototype outdoors, the users could perceive and navigate in the testing environments with very good accuracy. An overall performance analysis revealed an effectiveness of 88.85% for the SoV device. The effectiveness of a system is a measure that indicates the completion rate of the tasks with the system. The value was obtained considering all perception and mobility tasks performed by the users, where a task was considered successfully completed if the accuracy per task was higher than 85%. That is, the users were able to complete 88.85% of the tasks with an accuracy greater than 85%. The most difficult task for the users was to identify the dynamicity of objects (i.e., to count

the dynamic objects), especially in ego-static scenarios and when negative obstacles were also present Figure 7b. The users reported that, in these scenes, they mainly focused on the negative obstacles due to the powerful and distracting sound associated with it. Still, the perception of dynamic objects was the least trained aspect in the VTE and indoor sessions.

While the elevation of objects was not identified with maximum accuracy in perception tests, the users considered that avoiding hanging obstacles was easy to perform in mobility tasks. This feedback is also confirmed by users' performance results in task MT I-2, where all participants were 100% accurate in identifying and avoiding the tree branches.

The tests with special objects indicate that they could easily be perceived. The VI users could identify the presence of walls, holes in the ground and bus stop signs with 100% accuracy. They indicated the correct distance to such objects in 83% of the tests, and the correct direction in 91.6%.

With minimal training in using the SoV system outdoors, the users could perform the mobility tasks with very good accuracy. When comparing SoV with the white cane, we found out that, for our sample of rather inexperienced white cane users, mobility with the SoV device was accomplished with performance comparable to one with the white cane, and sometimes better. Analyzing the average walking speed over all users and all scenarios in which each modality was used revealed that, while using the white cane is the fastest, the SoV system has the advantage of reducing the average number of collisions.

User feedback was collected through questionnaires containing task specific questions, general questions about the system as well as individual user comments and suggestions. All items were designed to fit an answer format of a 5-point Likert-like scale, where 1 corresponds to strong disagreement and 5 corresponds to strong agreement. The feedback on how the system helps the users in accomplishing the tasks is summarized below as AVG (STD) values over all perception and mobility tests and all users:

T1—I found it easy to do this task with the device. Perception tests—4.39 (0.74), Mobility tests—4.70 (0.47)

T2—The device provides a good solution to problems I encounter in this task. Perception tests—4.54 (0.64), Mobility tests—4.80 (0.41)

T3—I am satisfied with the amount of time it took to complete this task. Perception tests—4.50 (0.75), Mobility tests—4.85 (0.37)

All users liked the device, for both perception and mobility. None of them believes it is unnecessarily complicated. They found it rather easy to use (to operate and switch between modes). They all disagree regarding the possible inconsistencies of the system and believe it works similarly in both the virtual training and testing environment and the real world. They are confident that most people would learn to use the device quickly. They don't find the device cumbersome to use, and although they are expecting design improvements for its commercial version, they were satisfied with the shape and functionality of the tested prototype. They were confident when using the device, and the confidence grew even more after the mobility tests. The visually impaired participants felt safe when using the device, and this feeling was even more emphasized for the mobility scenarios. Half of the participants thought that the device was comfortable, while the other two suggested several improvements. They all agreed that the SoV device would enhance their capacity for leisure activities, especially after going through the outdoor mobility sessions.

## 6. Conclusions

With minimal training in using the SoV system outdoor, the perception and mobility in the environments were achieved with very good accuracy. The tests revealed 88.85% effectiveness (task completion rate) of the SoV system. Regarding the perception, the system could be successfully used for perception (89.3% average accuracy) in noisy outdoor environments, without the environment sounds to have an obvious effect on the performance of perceiving the environment. The hear-through feature of the SoV headphones was found very useful by the users. Perceiving the dynamicity of obstacles can pose difficulties, especially in the presence of negative obstacles, which are signaled by the system with

a very powerful sound. In complex environments, the perception of individual obstacle elevation can pose difficulties to the visually impaired users. However, they all considered this to not be a major issue, since they can perceive the presence of these obstacles with the normal encoding and further have a distinctive feedback with the danger mode when approaching them closer than 1m. Even with a small amount of additional training time, perception of the environment improved from one testing session to a subsequent one.

With minimal training, the system could be successfully used in outdoor real-life environments to perform various mobility tasks. The visually impaired participants reported that performing mobility tasks with the SoV device was easier than building a detailed perception of complex scenes. They were also more satisfied about the solution provided by the SoV system for these tasks and by the time it took to complete them than for the perception tasks.

For inexperienced white cane users, mobility with the SoV device was accomplished with performance comparable to using the cane, and sometimes better. Users more experienced in using the white cane tend to rely more on the cane than on the system, when provided with both assistive devices. Less skilled white cane users chose to rely more on the SoV system. When using both modalities, users walk slower than only with the SoV system, as the amount of information coming from both devices can overwhelm users inexperienced with either of them.

The added value of SoV compared to the white cane was confirmed by the participants to consist in: providing early feedback about static and dynamic objects, providing feedback about elevated objects, walls, negative obstacles and signs.

**Supplementary Materials:** The following are available online at https://www.mdpi.com/article/10.3390/electronics10141619/s1.

**Author Contributions:** Conceptualization, S.C.; methodology, S.C. and R.-G.L.; software, O.Z., N.A.B. and S.C.; validation, S.C., A.B. and R.-G.L.; formal analysis, S.C.; investigation, A.B., S.C., O.Z., N.A.B. and R.-G.L.; data curation, O.Z., R.-G.L. and A.B.; writing—original draft preparation, O.Z.; writing—review and editing, all; supervision, S.C.; project administration, S.C.; funding acquisition, S.C. All authors have read and agreed to the published version of the manuscript.

**Funding:** This research was funded by the European Union's Horizon 2020 research and innovation program under grant agreement no 643636 "Sound of Vision" and by CNCS-UEFISCDI project PN-III-P2-2.1-PTE-2019-0810.

**Institutional Review Board Statement:** Rsearch ethics assent no. 13582/05.07.2016 for the activities of the project no. 643636 (H2020), Sound of Vision—Natural Sense of Vision Through Acoustics and Haptics, issued by the Research Ethics Committee of the "Gheorghe Asachi" Technical University of Iasi, Romania.

**Informed Consent Statement:** Informed consent was obtained from all subjects involved in the study.

**Conflicts of Interest:** The authors declare no conflict of interest.

## Abbreviations

The following abbreviations are used in this manuscript:

| | |
|---|---|
| SoV | Sound of Vision |
| VTE | Virtual Testing Environment |
| CT | Completion Time |
| TR | Training |
| PT | Perception Test |
| MT | Mobility Test |

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
