# Peer review of "Sensory Substitution for the Visually Impaired: A Study on the Usability of the Sound of Vision System in Outdoor Environments"

_electronics, doi:10.3390/electronics10141619_

Round 1

Reviewer 1 Report

This work deals with a delicate topic that is very attractive for the community. However, I have some major concerns that should be addressed.

The introduction is lacking a critical comparison between the different available technologies employed for VIP. A reader may think that the sound technology is the only alternative to the traditional white cane.
The literature is plenty of different studies employing for example camera-based systems.

- Kang, M.-C., et al. An enhanced obstacle avoidance method for the visually impaired using deformable grid. IEEE Trans. Consum. Electron. 2017, 63, 169–177.

- Yang, K.; Wang, K.; Hu, W.; Bai, J. Expanding the detection of traversable area with realsense for the visually impaired. Sensors 2016, 10, 1954.

Radar-based systems

- Di Mattia et al., "A feasibility study of a compact radar system for autonomous walking of blind people," 2016 IEEE 2nd International Forum on Research and Technologies for Society and Industry Leveraging a better tomorrow (RTSI), Bologna, Italy, 2016, pp. 1-5, doi: 10.1109/RTSI.2016.7740599.

- Cardillo, E.; Caddemi, A. Insight on Electronic Travel Aids for Visually Impaired People: A Review on the Electromagnetic Technology. Electronics 2019, 8, 1281. https://doi.org/10.3390/electronics8111281.

or infrared

- Bleau, M. et al., M. Blindness and the Reliability of Downwards Sensors to Avoid Obstacles: A Study with the EyeCane. Sensors 2021, 21, 2700. https://doi.org/10.3390/s21082700.

The novelty of this contribution, particularly compared to reference [3] is not clear and should be highlighted.

Some minor comments:

Letters should be used to separate the different subfigures. The figures’ captions should be self-explaining thus a more detailed description of the figure will be beneficial.

Some figures are not clear, e.g., Fig. 4 PT II-3.

It might be more interesting to know the energy consumption of the system to calculate the endurance. What is the weight of the battery used in the experiment?

Author Response

We thank the reviewer for the valuable comments and suggestions. We believe that the revised version of our paper is now considerably improved based on these suggestions, with respect to both content and shape. 

Reviewer 2 Report

Minor comments:
  - I recomend not to use Abbreviations in the Abstract. Define them the first time you write them in the regular text. Define all.
  - Change title of Sections to be more descriptive of what you describe inside.
  - Numbering of sections is also erroneous, i.e. 2.0.1.

Major comments:
  - Instead of writting the research questions (lines 64-71), it is mandatory to specify clearly the controbutions of the paper.It is very important you to differentiate justifiably the contributions of this paper infromt of the contributions of referenced papers [2] and [3] clarifying what was your role in the Sound of Vision (SoV) european Project [https://soundofvision.net/partners/].
  - Only 4 people has participated in the experiments, how did you select those people to be a representative of a wide statistical universe?
  - In Discussion Section, actually it is intended to discuss Graphics and data obtained and presented in the Experimental results Section. Please consider this in yur paper.

Reject Comments:
  - The structure and organization of the paper could be considerably improved, explaining first some parameters and a model of performance, then present experimental results and finally discussion of how the experimental results match theoretical model.
  - Experimental data could be better presented to avoid a lot of tipycal questions on validity of them.
  - I suppose that in prevoous work and the SoV european Project, the system was tested and work properly. So it is not understaood why you ask the question you ask in the Discussion Section. Probably yu must test the performance, not the working condition, of the system. Isn't it?

Author Response

(The authors gave the same response as above.)

Reviewer 3 Report

First off let me say that I found this to be a very interesting paper and I think with some minor fixes will be good to go. I will delineate.

1) Abstract - Minor point: The authors use "operational" environment but then switch to real world (and really mean real word). Drop the operational as there are researchers who do support operational environments (e.g., workplace, military, aviation industry, etc.). 

2) Abstract "response/reaction times" - Ok lets knock this one out now. If I remember Duncan Luce's book correctly neither reaction nor response times are what you were measuring. Also (as we will see in results) the time you collected was after N amount of tasks were done. So in my opinion you did a completion time. Seems like a minor thing but when the whole field of mental chronometry lives by reaction and response times I think calling it completion time would: 1) alleviate any potential confusion and 2) not make any difference in your findings

3) Overall comment: Your use of abbreviations (e.g., VIP, VTE, and RW) do not in any way contribute to helping the reader. I would only abbreviate a really long set of works that are not commonly used in the spoken language. Also since the visually impaired are the only participant population in your evaluation I believe you could take out VIP in every sentence you have it and by context the reader fully understands the group (visually impaired).

4) SoV - Ok I can get behind this one and I think it is quite a catchy name good job! But! Why did you not explain the phenomenology to the reader? I was really wanting to know 1) what/how the sounds were presented to the user and 2) same for the tactile. I think this would really help the reader.

5) Table 1 - You have TR and I do not think you said what TR stood for. Also, maybe I am just getting a bit long in the tooth but the font was a wee bit small and difficult to read.

6) Study design section - I would switch from calling it an experiment to an evaluation (similar to an HCI user evaluation) and that way you don't need to keep explaining away why it could not be an experiment. You did a very respectable job for a new system, with users, in the world.....be proud and explain your work.

7) Tables 2 and 3 - Again here PT and MT do not help in the least....I understand you used these for encoding for data collection and analysis but they did not help me the reader in understanding they just tripped me up.

8) Equation 1 - I was a bit lost here and did not understand your need to manipulate the data nor how you came up with the equation. For measuring time increments I am  more used to say log or ln of the data to normalize. 

9) Table 4 - See point 4 above....you did not explain how information is presented but then tease the reader with descriptions in this table. 

10) Figures - Overall too small and the light grey is impossible to read

11) Lines 412 - 416 -> I did not understand how/why this transformation came about....especially as you say it ".....was transformed in 1..." 1? Do you mean equation 1?

12) Final comment and lets go out on a high note. Thank you for a fantastic Discussion and Conclusions section! Well done!

Author Response

(The authors gave the same response as above.)

Round 2

Reviewer 1 Report

I thank the authors for effort put in addressing all my concerns.

Author Response

We thank the reviewer for all the valuable suggestions and input for improving the paper.

Reviewer 2 Report

Major comments:
  - Instead of writting the research questions (lines 64-71), it is mandatory to specify clearly the controbutions of the paper.It is very important you to differentiate justifiably the contributions of this paper infromt of the contributions of referenced papers [2] and [3] clarifying what was your role in the Sound of Vision (SoV) european Project [https://soundofvision.net/partners/]. SOME CLARIFYCATION IS STILL NEEDED. EXPOSED CONTRIBUTIONS ARE NOT REALLY CONTRIBUTIONS, but only characteristics of your system.
  - Only 4 people has participated in the experiments, how did you select those people to be a representative of a wide statistical universe? YOU MUST CLARIFY YOUR POINT IN THE PAPER IN ORDER TO SET BETTER THE UTILITY OF YOUR PAPER.
  - In Discussion Section, actually it is intended to discuss Graphics and data obtained and presented in the Experimental results Section. Please consider this in yur paper. NOT ENOUGTH.

Reject Comments:
  - Experimental data could be better presented to avoid a lot of tipycal questions on validity of them.
  - I suppose that in prevoous work and the SoV european Project, the system was tested and work properly. So it is not understaood why you ask the question you ask in the Discussion Section. Probably yu must test the performance, not the working condition, of the system. Isn't it? THE QUESTION IS THAT IN LINES 522-523. IF YOUR AIM IS TO SHOW THAT YOUR SYSTEM PRESENTS A GOOD USSUABILITY. IT IS STILL NOT CLEAR TO ME WHY YU DID NOT COMPARE TYOURS RESULTS WITH PREVIOUS RESULTS IN THE EUROPEAN PROJECT.

Author Response

We thank the reviewer for the clarifications. We believe that this second revision of the paper appropriately responds to the concerns.
